# Validation of Machine Learning Models for Craniofacial Growth Prediction

**DOI:** 10.3390/diagnostics13213369

**Published:** 2023-11-02

**Authors:** Eungyeong Kim, Yasuhiro Kuroda, Yoshiki Soeda, So Koizumi, Tetsutaro Yamaguchi

**Affiliations:** 1Department of Orthodontics, School of Dentistry, Kanagawa Dental University, Yokosuka 238-8580, Japan; koizumi@kdu.ac.jp (S.K.); t.yamaguchi@kdu.ac.jp (T.Y.); 2EDIAND Inc., Tokyo 135-0062, Japan; kuroda@ediand.co.jp (Y.K.); soeda@ediand.co.jp (Y.S.)

**Keywords:** craniofacial growth prediction, machine learning, longitudinal lateral cephalometric radiograph

## Abstract

This study identified the most accurate model for predicting longitudinal craniofacial growth in a Japanese population using statistical methods and machine learning. Longitudinal lateral cephalometric radiographs were collected from 59 children (27 boys and 32 girls) with no history of orthodontic treatment. Multiple regression analysis, least absolute shrinkage and selection operator, radial basis function network, multilayer perceptron, and gradient-boosted decision tree were used. The independent variables included 26 coordinated values of skeletal landmarks, 13 linear skeletal parameters, and 17 angular skeletal parameters in children ages 6 to 12 years. The dependent variables were the values of the 26 coordinated skeletal landmarks, 13 skeletal linear parameters, and 17 skeletal angular parameters at 13 years of age. The difference between the predicted and actual measured values was calculated using the root-mean-square error. The prediction model for craniofacial growth using the least absolute shrinkage and selection operator had the smallest average error for all values of skeletal landmarks, linear parameters, and angular parameters. The highest prediction accuracies when predicting skeletal linear and angular parameters for 13-year-olds were 97.87% and 94.45%, respectively. This model incorporates several independent variables and is useful for future orthodontic treatment because it can predict individual growth.

## 1. Introduction

Accurate diagnosis of a patient’s tooth placement, bite, and craniofacial development is necessary for selecting the appropriate orthodontic treatment [1,2,3]. This entails a thorough evaluation by an orthodontist to precisely identify the causes and crucial factors related to the issue [1,2]. Once the diagnosis is complete, an appropriate orthodontic appliance and treatment plan can be determined to achieve the best possible outcome for the patient [3]. Recently, among all patients undergoing orthodontic treatment, the proportion of adult patients is approximately 60% and is increasing with the rise in life expectancy and national income; however, the primary target population for orthodontic treatment includes pre-adolescent and adolescent patients [4]. Each patient’s occlusion, skeletal growth patterns, and profiles undergo significant changes during growth [5]. Therefore, predicting craniofacial growth in pediatric patients is important for proper treatment planning. Moreover, the treatment goal of orthodontics used to correct the bite cannot be determined if the craniofacial pattern of an individual cannot be predicted before, during, or after the treatment [6]. Craniofacial growth prediction is important not only in planning and providing treatment to patients but also in assessing treatment prognosis [2,7].

In recent years, technological advances have allowed artificial intelligence (AI) to make revolutionary advances in the field of medicine [8,9]. AI is a subfield of computer science involving the development of programs with computers that recognize and reason information and ultimately convert that information into intelligent actions [10]. AI is very broad and includes various disciplines such as reasoning, natural language processing, planning, machine learning (ML), and deep learning (DL) [11].

Currently, ML is the most commonly used AI application in medical and dental fields, and various AI and ML integration programs have been developed to assist orthodontists in clinical practice, including diagnosis, treatment planning, and evaluation of treatment results and growth [12,13]. Xie et al. [14] used an artificial neural network (ANN) system to determine whether extraction or non-extraction treatments were optimal before orthodontic treatment and reported that the ANN worked with 80% accuracy. Lee et al. [15] developed a new framework for finding cephalometric landmarks using a Bayesian convolutional neural network focusing on detecting cephalometric landmarks. AI has also automatically identified and classified skeletal malocclusions from three-dimensional cone beam computed tomography craniofacial images [16]. Therefore, the use of AI and ML to determine the need for orthodontic treatment improves the diagnostic accuracy [17]. ML algorithms applied to orthodontic specialties currently and in the future offer promising tools for improving clinical practices [18].

Predicting craniofacial growth is one of the most challenging issues in orthodontic treatment. Among the studies on craniofacial growth prediction to date, many studies have reported on the observation of craniofacial growth using lateral cephalometric radiographs. These methods include mesh diagrams [19,20], grids [21], templates [22], the Ricketts visual treatment objective [23,24], and multilevel models [25,26,27,28,29]. These growth prediction methods estimate a patient’s residual growth based on average annual increments. However, these approaches do not account for individual differences, and the average annual growth rate applies to all patients. Manabe et al. [30] cited the inability of individuals to follow continuing growth as a limiting factor in their study and stated that, ideally, a longitudinal study would be desirable.

Morphometric analysis of longitudinal-lateral cephalometric radiographs provides numerical values for the amount and direction of individual growth. In this study, we aimed to identify the best model for predicting the accuracy of four ML models, including one multiple regression analysis (MRA) and three ML models that had not been previously validated, using longitudinal lateral cephalometric radiographs to predict longitudinal craniofacial growth in the Japanese population.

This study not only validated the methods reported in previous studies but also presented new machine learning methods that have not been validated before. By validating the craniofacial growth prediction model using the various methods, new insights can be gained. Additionally, we focus on the prediction of long-term craniofacial growth without orthodontic intervention in the Japanese population. Craniofacial growth patterns may differ by region and race [31].

## 2. Materials and Methods

### 2.1. Research Materials

In this study, we analyzed longitudinal data related to craniofacial growth and development obtained from the Department of Orthodontics, School of Dentistry, Kanagawa Dental University, Japan. These lateral cephalometric radiographs were obtained annually from students enrolled at Shioiri Elementary School between 1965 and 1973. We followed the Declaration of Helsinki guidelines, and this study was approved by the Kanagawa Dental University Research Ethics Review Committee (approval number: 646; date of approval: 12 March 2020). The study included 59 individuals (27 boys and 32 girls) without orthodontic treatment whose lateral cephalometric radiographs were of analyzable quality. Additionally, these individuals had complete longitudinal records for ages 6, 12, and 13.

### 2.2. Data Collection

Lateral cephalometric radiographs were digitized to their actual size for computer-based craniofacial analysis using EPSON ES-2200 V. 3.070 (Seiko Epson Corporation, Nagano, Japan). A single observer performed all tracings of the lateral cephalometric radiographs using PowerPoint V. 16.66.1(22101101) (Microsoft Co., Redmond, WA, USA). Moreover, to analyze the growth process over 7 years (ages 6 to 13), each patient’s age-specific lateral cephalometric radiographs were superimposed using the sella as the reference point, ensuring that the anterior cranial base (S–N plane) maintained the same angle.

We identified 26 skeletal landmarks commonly used in orthodontic diagnosis: the sella, porion, basion, nasion, orbitale, A point, pogonion, B point, PNS, ANS, R1, R3, articulare, menton, mx1-incisor superius, mx1 root, md1-incisor inferius, md1 root, occlusal plane, mx6 distal, mx6 root, md6 distal, md6 root, gnathion, gonion, and condylion (Figure 1). Using the ImageJ software V. 1.53t (National Institutes of Health, Bethesda, MD, USA), we plotted these landmarks on overlaid tracings of lateral cephalometric radiographs and detected the coordinates of each landmark.

In this study, we examined two types of predictions based on the collected coordinate values. The first prediction forecasts the x- and y-coordinates of the landmarks at the age of 13 using their values from ages 6 to 12, and the second prediction involved distances (Table 1, Figure 2) and angles (Table 2, Figure 3) typically used in orthodontic treatments. We used values from ages 6 to 12 as inputs to predict their respective values at age 13.

### 2.3. Data Normalization and Validation

#### 2.3.1. Interpolation of Missing Values

When analyzing the lateral cephalometric radiographs obtained annually, we encountered four instances of missing data for ages other than 6, 12, and 13 for unspecified reasons. We interpolated the values from temporally adjacent data points for these missing data points to create and use provisional values for analysis.

#### 2.3.2. Normalization of Coordinate Values

Considering that the coordinate values detected using Image J (National Institutes of Health, Bethesda, MD, USA) lacked precise alignment in position or inclination across the subjects before conducting the analysis, we normalized the coordinates using the following procedure: First, we set the sella as the origin (0, 0). Second, we rotated the coordinates such that the N–S plane was aligned parallel to the *x*-axis. Third, we converted the coordinates from pixels to millimeters. We applied the same conversion coefficient to all images because they were captured and scanned at the same scale.

#### 2.3.3. Tested Statistical Methods

To predict craniofacial growth, we considered the limited number of available samples; the fact that growth at each measurement point is not necessarily linear in the coordinates centered on the sella; and the uncertainty regarding which measurement points, distinctive distances, and angles significantly affect future growth predictions. Consequently, we compared multiple statistical methods. Considering the multiple measurement points and limited sample size, we selected five methods. For linear modeling with variable selection features, we used the stepwise method for MRA and the least absolute shrinkage and selection operator (LASSO). We employed a radial basis function network (RBFN) and multilayer perceptron (MLP) as nonlinear modeling methods. Additionally, we incorporated a gradient-boosted decision tree (GBDT) as a tree-based method.

The following sections describe the modeling procedure for each method. For simplicity, the x and y coordinates of the measurement points as well as the lengths and angles commonly used in orthodontic treatment are collectively referred to as “measurement values” in the subsequent sections.

MRA using the stepwise method

MRA is based on a simple linear model for growth prediction. We used all measurement values from 6 to 12 years as independent and at 13 years as dependent variables. $X_{i}\;and\;Y_{j}$ are the independent and dependent variables, respectively, where *i* and *j* are subscripts indicating the measurement points. Denoting $X_{i}$ as an independent variable and $Y_{j}$ as a dependent variable, we formulated the following equation:(1)$$Y_{j}=A_{j,0}+\sum_{i=1}A_{j,i}X_{i},$$
where $A_{j,i}$ represents the coefficient indicating the influence of each independent variable.

We used the least squares method to determine each $A_{i}$ such that
(2)$$\sum_{n=1}^{N}{\left(Y_{j,n}^{\prime }-Y_{j,n}\right)}^{2},$$
is minimized. Here, *n* represents the index of the training data, *N* is the total number of training data points, and ${Y^{\prime }}_{j,n}$ is the actual measured value at 13 years. Additionally, in implementing optimization using the least-squares method, some $A_{i}$ values were used, where the stepwise method [32] selected variables. In the least squares method, the stepwise method evaluates the model using the adjusted R^2^ [33] and adopts the independent variables that provide the best evaluation. This utilization of adjusted R-squared prevents the inclusion of an excessive number of independent variables, thus promoting a more robust and parsimonious model. The stepwise method was performed by the forward-backward method. Starting with no independent variables selected, if adding an independent variable improves the adjusted R-squared, that variable is added. On the other hand, if an independent variable has already been selected, and removing an independent variable improves the adjusted R-squared, the variable is repeatedly removed until there is no change, and then the independent variable is selected.

LASSO

LASSO [34] represents a form of linear regression that maintains the same relationship between independent and dependent variables, as expressed in Equation (1), where n represents the index of the training data. *λ* is constant to control the regularization by the L1 term. The size of *λ* is determined by performing cross-validation for each measurement point and selecting the best result.

The optimization method distinguishes LASSO from multiple regression analysis. LASSO employs the following equation, incorporating L1 regularization:(3)$$\sum_{n=1}^{N}{\left(Y_{j,n}^{\prime }-Y_{j,n}\right)}^{2}+\lambda \sum_{i}\left|A_{j,i}\right|,$$

This approach constrains the magnitude of the coefficient $A_{i}$ while minimizing prediction errors. Such regularization ensures that the coefficients remain moderate, thereby preventing overfitting. Furthermore, the nonessential coefficients for prediction are naturally driven toward zero, facilitating the automatic selection of independent variables.

RBFN

RBFN [35] models the dependent variable $Y_{j}$ as a linear combination of the base function $f_{i}\left(X\right)$ with respect to independent variable $X_{i}$. *k* and *j* are subscripts indicating the base function and which measurement point is the dependent variable, respectively.

Utilizing non-linear functions as base functions allows the approximation of any non-linear function considering an adequate number of clusters.
(4)$$Y_{j}=\sum_{k=1}^{M}w_{j,k}f_{k}(X_{0},X_{1},\ldots ),$$
where $w_{j,k}$ denotes the weight of the linear combination of base functions. *i*, *j*, and *k* are subscripts indicating which measurement point is the independent variable, dependent variable, and base function, respectively. *σ_k_* is the standard deviation of the base function for *k*.

We used a Gaussian function as the basis function, which can be expressed as follows:(5)$$f_{k}\left(X_{0},X_{1},\ldots \right)=\exp\left(-\frac{{\sum_{i}\left(X_{i}-a_{k,i}\right)}^{2}}{2\sigma_{k}^{2}}\right).$$

The Gaussian function parameters $a_{k,i}$ are derived from the centroid coordinates of the clusters defined by the fuzzy c-means method [36]. Weights $w_{j,k}$ were optimized to minimize the discrepancy between the predicted and actual dependent variables. *j* is a subscript indicating which measurement point is the dependent variable, and n represents the index of the training data, N is the total number of training data points.
(6)$$\sum_{n=1}^{N}{\left(Y_{j,n}^{\prime }-Y_{j,n}\right)}^{2}$$

The number of clusters in the fuzzy c-means and the standard deviation of the Gaussian function are the hyperparameters for the RBFN-based model. We employed Optuna [37], a Bayesian optimization package, to search for optimal values. The search range for the number of clusters was set from 2 to 20, and *σ* was set from 0.1 to 10. Momentum stochastic gradient descent (momentum SGD) is used when optimizing $w_{j,k}$, and a learning rate of 0.0001 and a momentum of 0.9 were used.

MLP

By combining multiple simple perceptrons, MLPs [38] can approximate a nonlinear function. The following formula represents a simple perceptron:(7)$$z_{k}=act\left(w_{k,0}+\sum_{i}w_{k,i}X_{i}\right)$$

In this formula, “act” is a nonlinear activation function. $w_{k,i}$ denotes the connection weight from the *i*th input to the *k*th output, and $w_{k,0}$ refers to the bias. The MLP structures are stacked multiple times. *i*, *k*, $X_{i},\;and\;z_{k}$ are the indexes representing each node of the perceptron, output node, input of the perceptron, and the output of the perceptron, respectively. When an MLP with a single intermediate layer is used, the model describes the dependent variable as follows:(8)$$Y_{j}=act\left(v_{j,0}+\sum_{k}w_{j,k}z_{k}\right)=act\left(v_{j,0}+\sum_{k}v_{j,k}act\left(w_{k,0}+\sum_{i}w_{k,i}X_{i}\right)\right),$$
where $v_{j,k}$ represents the connection weight from the *k*th input to the jth output, and $v_{j,0}$ is the bias. *i*, *j*, and *k* are the indexes representing which measurement point is the independent variable, dependent variable, and the node number of the MLP’s middle layer, respectively.

In this study, we used an MLP with two intermediate layers comprised of 64 nodes and selected ReLU [39] as the activation function.

The MLP connection weights were trained using the momentum stochastic gradient descent method [40] to minimize the error between the predicted and measured dependent variable values. We set the momentum at 0.9 and the learning rate coefficient at 0.01.

GBDT

The GBDT [41] combines multiple decision trees to make predictions. It uses decision trees to represent weak learners. Based on the given independent variable values, it constructs decision trees and creates another tree using the prediction errors from the previous tree. After repeatedly performing this process, the results from numerous decision trees are assembled to make predictions. The GBDT maintains high accuracy even when unnecessary independent variables are used, and its versatility has led to its widespread use in tabular data predictions in recent years. In this study, we used LightGBM [42], an algorithm based on GBDT, for training and evaluation. We employed the LightGBM Tuner from Optuna [37] (Preferred Networks Inc., Tokyo, Japan) to optimize hyperparameters such as the learning rate coefficient.

#### 2.3.4. Cross-Validation

For evaluation, we divided the dataset of 59 individuals into ten subsets and employed cross-validation. We used nine of these subsets to train our statistical and ML models and reserved one for validation purposes. We repeated this process with different subsets serving as the validation set, resulting in ten unique combinations. We averaged the results from all evaluations to derive the final evaluation metric.

#### 2.3.5. Validation using LASSO Age-Specific Input Data

In clinical practice, predicting craniofacial growth from a single point in a patient’s early development holds great promise for therapeutic treatment [43]. Wood et al. [43] compared *Y*-axis prediction accuracy at post-adolescent stages with data from younger ages alone and data from younger and older ages combined. In our study, as in Wood et al. [43], we used data for only 6-year-olds and combined data for 6-year-olds and older adults to examine prediction accuracy in order to make comparisons.

To predict the coordinate values of skeletal landmarks, skeletal linear parameters, and angular parameters for age 13 using the LASSO predictive model, we used data for age 6 only; data for ages 6 to 7, 6 to 8, 6 to 9, 6 to 10, 6 to 11, and 6 to 12 were used as input data.

### 2.4. Statistical Analysis

We employed the root mean square (RMS) to compare the gap between predicted and actual coordinates at age 13 when generating predictions based on measurement points. For predictions, we compared the RMS of the difference between the predicted and actual values of these measures at age 13 using standard lengths and angles, which are customary in orthodontic treatment. Accuracy was calculated as follows:Accuracy (%) = 100 × (1 × (RMSE/AVERAGE of the actual values)).(9)

We calculated the root-mean-square error (RMSE) as the average error between the predicted and actual values for the age of 13 years.

## 3. Results

The prediction model using MRA had an average error of 2.49 mm for the coordinate value of skeletal landmarks (Table 3), 2.45 mm for skeletal linear parameters (Table 4), and 4.31° for skeletal angular parameters (Table 5). The prediction model using LASSO had an average error of 1.41 mm for the coordinate value of skeletal landmarks (Table 3), 1.49 mm for skeletal linear parameters (Table 4), and 1.94° for skeletal angular parameters (Table 5). The prediction model using RBFN had an average error of 8.34 mm for the coordinate value of skeletal landmarks (Table 3), 4.81 mm for skeletal linear parameters (Table 4), and 6.24° for skeletal angular parameters (Table 5). The prediction model using MLP had an average error of 4.66 mm for the coordinate value of skeletal landmarks (Table 3), 3.29 mm for the skeletal linear parameters (Table 4), and 4.44° for the skeletal angular parameters (Table 5). The prediction model using the GBDT had an average error of 3.43 mm for the coordinate value of skeletal landmarks (Table 3), 2.24 mm for skeletal linear parameters (Table 4), and 3.02° for skeletal angular parameters (Table 5).

Among all prediction models examined, those using LASSO had the lowest average error for all values. The prediction accuracies with linear and angular skeletal parameters were 97.87% (Table 4) and 94.45% (Table 5), respectively.

Although LASSO had the best accuracy among all the methods considered, to elucidate its prediction accuracy in more detail, we verified the prediction by age (Figure 4). The average error was 3.28 mm when only the data of 6-year-olds were entered, and 1.41 mm with all data from those ages 6 to 12 years.

## 4. Discussion

Craniofacial growth is a fundamental topic in orthodontics with a long history of research. Craniofacial growth prediction, particularly in clinical practice, is useful for diagnosis and treatment planning [2,7]. Therefore, various methods have been developed to predict the craniofacial growth. In the modern era, obtaining lateral cephalometric radiographs of patients who have not received orthodontic treatment solely for research purposes is ethically difficult. Consequently, only a few studies have tracked the longitudinal craniofacial growth of individuals, and individual patient predictions have not reached the accuracy level required in clinical practice [30,44]. Meanwhile, AI has advanced in numerous fields in recent years, including medicine. ML, a branch of AI, can improve the accuracy of diagnosis in orthodontic treatment [17]. Lateral cephalometric radiographs were used to track the longitudinal craniofacial growth in individuals not undergoing orthodontic treatment. We then examined the prediction model of craniofacial growth using MRA [45,46,47,48,49], a statistical method used in conventional methods of craniofacial growth prediction, and ML, which has the potential for development in many fields. We aimed to identify the best model for prediction accuracy. The results revealed that LASSO had the smallest average error and high accuracy for all measurement items, including the coordinate values of skeletal landmarks and linear and angular parameters.

Mathematical models using various statistical methods have been examined to predict craniofacial growth [25,29,45,46,47,48,49]. These were examined using multilevel models (MLMs) and MRA mathematical models. Two studies developed mathematical models from the MLM [25,29]. Buschang et al. [29] developed a growth prediction model incorporating the mean annual velocities from MLMs using randomly selected lateral cephalometric radiographs of nonidentical subjects. The model was compared to a polynomial model of the population growth curve using measurements at ages 11, 12, and 13 to predict the sella-gnathion at age 15. In their study, growth predictions for boys and girls were 76 to 77% accurate. The sella-gnathion in our study had an accuracy range of 93.26% to 98.39%. Meanwhile, Chvatal et al. [25] aimed to determine whether additional longitudinal data would increase the prediction accuracy of craniofacial growth in their participants. Using serial lateral cephalometric radiographs of the same patient ages 6 to 15, we developed longitudinal growth curves that could accurately model individual differences and a prediction model for craniofacial growth from MLM. They predicted sella-menton, and the prediction accuracy of the model was approximately 83 to 90%. The sella-menton was not considered in the present study. Chvatal et al. [25] determined that longitudinal data did not significantly improve the prediction accuracy of the measurements used in their study. However, they stated that more longitudinal data could improve prediction accuracy. Another study developed mathematical models using MRA to predict mandibular growth [45,47]. Oueis et al. [45] aimed to predict mandibular growth in healthy Japanese children ages 4 and 9 in the same patients. Mathematical models were developed for MRA using lateral cephalometric radiographs, which predicted the nasion-sella-gnathion and sella-gnathion with an accuracy of 72% and 61% for the direction and amount of growth, respectively. Subsequently, Yano et al. [47] set their participants as Japanese children with an anterior crossbite at 4 and 9 years in the same patient. Using lateral cephalometric radiographs, mathematical models were developed from MRA to predict mandibular growth. The developed mandibular growth predicted the nasion-sella and sella-gnathion with 68% and 69% accuracy for the direction and amount of growth, respectively. The accuracy of the nasion-sella-gnathion and sella-gnathion using the MRA prediction model in our study was 99.28% and 97.94%, respectively, and the accuracy range for all methods was 94.62% to 99.71% and 93.26% to 98.39%, respectively.

By contrast, other studies incorporated bone maturity and developed mathematical models using MRA [46,48,49]. Sato et al. [48] collected lateral cephalometric and hand-wrist radiographs of nonidentical female patients (initial and final ages of 8.3 and 18.4, respectively) for their study. A growth prediction model was developed using MRA to predict mandibular growth (gnathion-condylion) based on skeletal maturity indicators assessed using hand-wrist radiographs. To validate the developed growth prediction model, they compared the difference between predicted and actual measurements in another 22 female patients (mean initial and final ages of 10.8 and 18.6, respectively). The results revealed that the average error of the gnathion-condylion was 4.3 to 4.9 mm. Mito et al. [49] developed a model to predict mandibular growth (gnathion-condylion) from MRA according to skeletal maturity indicators assessed by the cervical spine and evaluated from lateral cephalometric radiographs of female patients who were not the same participants. The average error of the gnathion-condylion was 3.48 ± 1.97 mm. With an average error of 2.72 mm for the gnathion-condylion, our results are better. However, this method was derived from lateral cephalometric radiographs of children at various stages of growth (7 years to 13 years), making it difficult to apply it to the prediction of each affected child.

Sato et al. [48] and Mito et al. [49] used non-same-subject study material. However, studies incorporating bone maturity and serial lateral cephalometric radiographs have also been conducted. Moshfeghi et al. [46] used serial lateral cephalometric radiographs of 33 Iranian girls ages 9 to 11 from the same patient. They developed a model to predict mandibular growth (articular pogonion) based on skeletal maturity indicators assessed in the cervical spine using MRA. The prediction model for mandibular growth developed by Moshfeghi et al. [46] was compared with that developed by Mito et al. [49]. The average errors of the methods proposed by Moshfeghi et al. [46] and Mito et al. [49] are 0.149 mm and 5.87 mm, respectively. Moshfeghi et al. [46] attributed the better results to the ability of longitudinal lateral cephalometric radiographs to predict individual growth. The articular-pogonion was not considered in the present study.

As demonstrated in these studies, MLM and MRA have been used to predict craniofacial growth. Although not as accurate, these studies demonstrate the importance of statistical and mathematical analyses in predicting craniofacial growth. However, the application of statistical methods, such as MLM and MRA, as prediction models may be too simple to predict complex craniofacial growth because they are based on the linear analysis of independent variables.

By contrast, ML uses various sample data to generate mathematical models that generalize and predict specific patterns. This feature may be useful in predicting complex craniofacial growth in humans [50]. Jiwa et al. [51] developed a growth prediction model from a DL algorithm with 17 mandibular landmarks using serial lateral cephalometric radiographs of growing patients. The accuracy of this model was compared to Ricketts’s growth prediction [52], although the prediction error was larger and less accurate than Ricketts’s growth prediction. They suggested increasing the data volume and training to improve the accuracy of their model. Zhang et al. [53] used lateral cephalometric radiographs of the same patients who had an anterior crossbite, with a mean age of 8 to 14. They developed a deep convolutional neural network model that automatically predicts normal mandibular growth or overgrowth. The SNB, ANB, FMA, sella-nasion-pogonion, nasion-sella-gnathion, nasion-sella-articulare, articulare-gonion-menton, articulare-gnathion, gnathion-condylion, gonion-gnathion, and condylion-gonion were measured. The deep convolutional neural network model demonstrated a prediction accuracy of approximately 85%. Moon et al. [54] used serial lateral cephalometric radiographs of the same patients, with a mean age ranging from 10.9 to 14.2. A growth prediction model was developed from a partial least squares algorithm using the coordinates of 78 lateral cephalometric landmarks. The results suggest that the younger the age at the time of prediction, the larger the prediction error (0.03 mm/year), and the older the age, the more accurate the prediction results. Wood et al. [43] investigated the pre-pubertal (mean age of 11 to 12), pubertal (mean age of 13 to 14), and post-pubertal (mean age of 15 to 16) stages of the same male patients. Longitudinal lateral cephalometric radiographs obtained at three time points were used as research materials. These were then used to develop a growth prediction model using various ML models, including LASSO. They aimed to predict mandibular length (gnathion-condylion) and *Y*-axis (nasion-sella-gnathion) post-puberty in males. The growth prediction model developed from each ML was used to compare differences between the predicted and actual measurements. Compared to our study, the patients, observation, and prediction period were different, although the similarities are that longitudinal lateral cephalometric radiographs can be used to determine individual growth using the study data. Their study and our study included 39 and 56 independent variables and 163 and 468 lateral cephalometric radiographs, respectively. The ML model validated in the study by Wood et al. [43] as well as in ours was LASSO. The prediction accuracy of the mandibular length (gnathion-condylion) validated by LASSO in the study by Wood et al. [43] ranged from 97.18% to 97.46%, whereas our study identified an accuracy of 98.26%. The prediction accuracy of the *Y*-axis ranged from 97.88% to 98.34% in the study by Wood et al. [43], and the accuracy in our study was 99.28%. Our results demonstrated higher accuracy than those of Wood et al. [43] for both mandibular length (gnathion-condylion) and the *Y*-axis (nasion-sella-gnathion). The better accuracy obtained in our study may be attributed to our larger sample size than that of Wood et al. [43] and the use of lateral cephalometric radiographs obtained at yearly intervals over an 8-year period.

Periodic and serial lateral cephalometric radiographs predict pure craniofacial growth in the absence of orthodontic treatment [55]. However, performing periodic and serial lateral cephalometric radiography in patients who have not received orthodontic treatments is ethically difficult. Therefore, many studies on craniofacial growth prediction have been conducted using noncontinuous lateral cephalometric radiographs obtained at various growth stages [29,48,49]. However, these studies are not suitable for individual predictions. This study included 59 boys and girls between the ages of 6 and 13 and obtained eight serial lateral cephalometric radiographs at 1-year intervals. Our prediction model for craniofacial growth demonstrated better accuracy than those developed in the various growth prediction studies described above. Our study used more periodic and serial data than the growth prediction studies using lateral cephalometric radiographs described above. Therefore, we obtained more accurate growth predictions by monitoring individual growth.

In our study, LASSO had the highest accuracy among all methods considered. Furthermore, to elucidate the accuracy of the LASSO predictions, we examined the predictions by age (Figure 4). The accuracy improved with increasing age. Moon et al. [54] concluded that the prediction error of the partial least squares prediction model increases with age at the time of prediction and that the older the age, the more accurate the prediction results. Wood et al. [43] also compared the accuracy of predicting the *Y*-axis in the post-pubertal stage with data for lower ages only and with data for lower and higher ages combined. The results revealed that the prediction accuracy improved with increasing age. Our study replicates the trends observed in both studies.

However, whether LASSO achieves the highest accuracy among various ML methods must be considered from a statistical perspective. We used 26 coordinated values of skeletal landmarks, 13 skeletal linear parameters, and 17 skeletal angular parameters as independent variables. Leslie et al. [56] emphasized that MRA alone can only make limited predictions because it is exploratory only when a large number of independent variables are affected. Zhang et al. [53] also noted that the application of regression equations to predict mandibular growth is based on an analysis of the linear combination of covariates. Therefore, predicting the complex growth of the mandible may be overly simple. MRA models the relationships among the data from a linear perspective. However, because craniofacial growth patterns often have nonlinear properties, adequately capturing all properties of the data in this manner can be difficult. However, the use of AI can alleviate this concern. The results of this study demonstrate that certain features can be detected by the algorithm regardless of the method used, even when too many variables are used as the input data. MRA did not properly capture the features, and it is thought that variable selection could not be done efficiently.

LASSO [57] is a type of linear regression aiming at feature selection and model simplification, allowing the model to remove unwanted features and prevent overfitting. LASSO is a penalized parametric regression model performing feature selection. This method adds a penalty to the absolute values of the regression coefficients, reducing the influence of irrelevant features (i.e., explanatory variables that are not important for the target variable). This property is advantageous for noisy data and data with multicollinearity and improves the prediction accuracy of the model. LASSO is also useful in preventing overfitting by appropriately setting penalty terms [41]. This is particularly useful when the number of features is large or when the number of samples is small. By preventing overfitting, the model can provide more consistent predictions for new data. Additionally, LASSO produces sparse solutions, making predictions more efficient, even when the number of measurements is large. Certain features may be key factors in predicting craniofacial growth, explaining why LASSO achieved the highest accuracy.

RBFN is a type of neural network particularly effective when handling high-dimensional nonlinear data [35]. However, the results of our study indicate that RBFN has large errors, which may be caused by the model causing overfitting [41]. MLP can capture complex patterns in nonlinear datasets [38]. However, similar to RBFN, MLP is prone to overfitting, and the optimal network structure is difficult to determine. This may be one reason why the prediction accuracy of MLP was lower than that of the other methods.

GBDT [41] is a method for creating prediction models by combining several decision trees, with each individual decision tree acting as a weak learner enabling weak prediction models to be combined, creating strong models. However, GBDT also requires the adjustment of hyperparameters and may cause overfitting. LASSO demonstrated better results owing to its ability to select features and suppress overfitting.

In conclusion, LASSO is the most suitable prediction model for craniofacial growth. However, the optimal model varies depending on the data and learning algorithms, and these methods should be reevaluated for different learning algorithms and new datasets [58]. Moreover, as human growth patterns are strongly influenced by individual genetic and environmental factors, these factors must be considered when creating models.

The results of our study showed that the error values for some measurements were larger than for others. The error values for Occlusal_pl_to_SN were very large for all models except GBDT (Table 5). The position of the premolars and molars determines Occlusal_pl_to_SN. Between 6 and 13 years, there is active permanent tooth eruption and craniofacial growth that considerably changes tooth positioning [59]. Various factors that may be involved in craniofacial growth (cranial base flexion, eruption, vertical dimension, occlusal_pl_to_SN, intrinsic maxillary and mandibular growth, genetics, environment, and so on) are discussed, and their interactions, especially the importance of the Occlusal_pl_to_SN slope as a major determinant in establishing the mandibular position, is not fully understood, making prediction of Occlusal_pl_to_SN difficult [60]. Therefore, it is likely that the GBDT, with its high representativeness, was relatively successful in its predictions, while other models were not.

Also, the ANB and porion error values of GBDT were very large (Table 3). We report that the specific cause of large porion and ANB error values caused by GBDT is unclear. Future work may require additional research and adjustments to the data and model.

The strength of our study is that the growth observation period for all participants was constant. The method of interpreting growth may vary depending on the measurement method applied and observation interval. In our study, as the growth observation intervals were predetermined, no errors in growth prediction occurred during the non-constant growth observation periods. However, our study has some limitations. First, we could not account for the effects of age-related growth differences. Therefore, additional variables reflecting skeletal age should be included. Second, the genetic and clinical characteristics of the patients and other factors were not included in the algorithm, and the performance of the growth prediction model could be improved by adding these factors.

In the current study, a sophisticated statistical approach called ML was applied to validate the craniofacial growth prediction model. As growth is a complex process influenced by genetic and environmental factors and varies by sex and race, ML growth prediction models remain imperfect. However, rapid advances in AI and continuous improvements in data quality and quantity can serve as guidelines for approximate growth predictions. Therefore, ML-based growth prediction models are expected to become reliable clinical support systems for orthodontists in the near future.

## 5. Conclusions

Among the statistical and ML methods used to predict longitudinal craniofacial growth in the Japanese population, LASSO had the high prediction accuracy for all linear and angular skeletal parameters. LASSO is the most effective method for solving the problems of feature selection and overfitting when constructing a craniofacial growth prediction model.

## Figures and Tables

**Figure 1 diagnostics-13-03369-f001:**
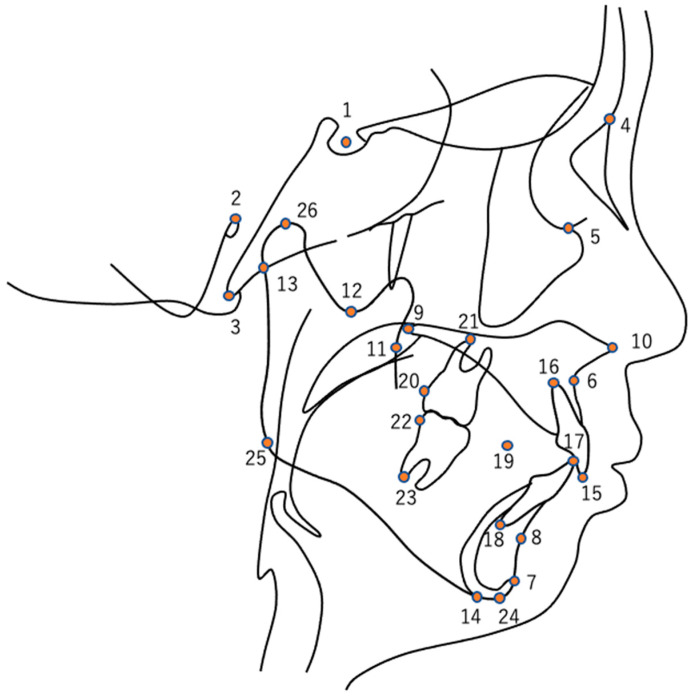
Skeletal landmarks: (1) sella; (2) porion; (3) basion; (4) nasion; (5) orbitale; (6) A point; (7) pogonion; (8) B point; (9) PNS; (10) ANS; (11) R1; (12) R3; (13) articulare; (14) menton; (15) mx1-incisor superius; (16) mx1 root; (17) md1-incisor inferius; (18) md1 root; (19) occlusal plane; (20) mx6 distal; (21) mx6 root; (22) md6 distal; (23) md6 root, (24) gnathion; (25) gonion; and (26) condylion.

**Figure 2 diagnostics-13-03369-f002:**
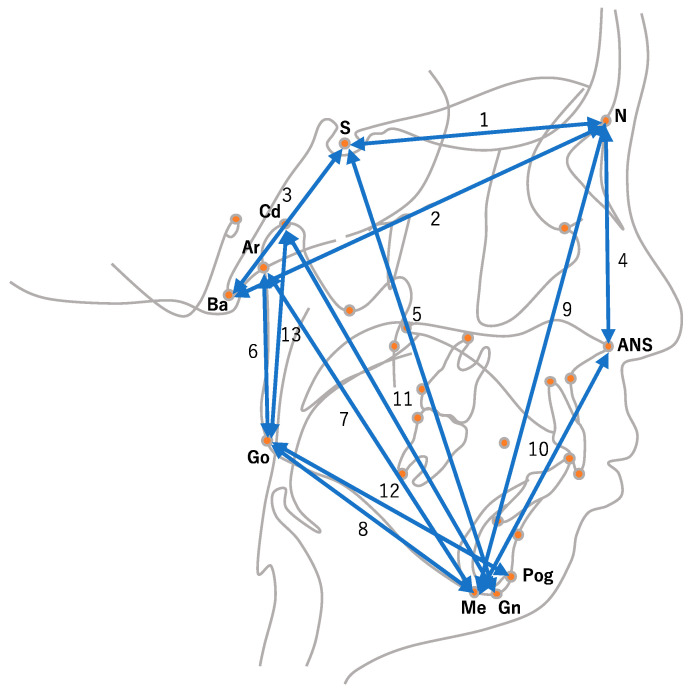
Skeletal linear parameters: (1) N-S; (2) N-Ba; (3) S-Ba; (4) N-ANS; (5) S-Gn; (6) Ar-Go; (7) Ar-Me; (8) Go-Me; (9) N-Me; (10) ANS-Me; (11) Gn-Cd; (12) Pog-Go; (13) Cd-Go.

**Figure 3 diagnostics-13-03369-f003:**
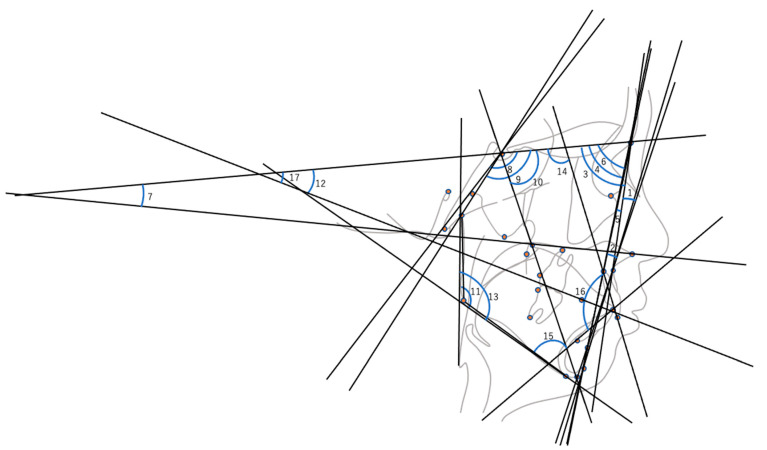
Skeletal angular parameters: (1) convexity; (2) A–B plane; (3) SNA; (4) SNB; (5) ANB; (6) N-Pog to SN; (7) nasal floor to SN; (8) NSBa; (9) NSAr; (10) NSGn; (11) ArGoMe; (12) mandibular pl to SN; (13) gonial angle; (14) U1 to SN; (15) L1 to mandibular pl; (16) interincisal angle; (17) occlusal pl to SN.

**Figure 4 diagnostics-13-03369-f004:**
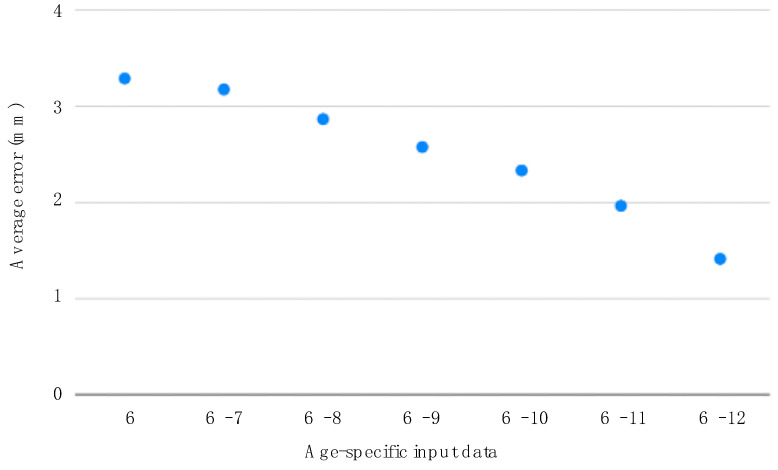
Average error due to limit on number of age-specific input data for the least absolute shrinkage and selection operator prediction model (mm).

**Table 1 diagnostics-13-03369-t001:** Standard skeletal linear parameters commonly used in orthodontic treatment.

Skeletal Linear Parameters	Landmarks to Be Used
N–S	(Nasion, Sella)
N–Ba	(Nasion, Basion)
S–Ba	(Sella, Basion)
N–ANS	(Nasion, ANS)
S–Gn	(Sella, Gnathion)
Ar–Go	(Articulare, Gonion)
Ar–Me	(Articulare, Menton)
Go–Me	(Gonion, Menton)
N–Me	(Nasion, Menton)
ANS–Me	(ANS, Menton)
Gn–Cd	(Gnathion, Condylion)
Pog–Go	(Pogonion, Gonion)
Cd–Go	(Condylion, Gonion)

**Table 2 diagnostics-13-03369-t002:** Standard skeletal angular parameters commonly used in orthodontic treatment.

Skeletal Angular Parameters	Landmarks to Be Used
Convexity	(Nasion, A point), (A point, Pogonion)
A–B plane	(A point, B point), (Nasion, Pogonion)
SNA	(Sella, Nasion), (Nasion, A point)
SNB	(Sella, Nasion), (Nasion, B point)
ANB	(Nasion, A point), (Nasion, B point)
N-Pog to SN	(Nasion, Pogonion), (Sella, Nasion)
Nasal floor to SN	(ANS, PNS), (Sella, Nasion)
NSBa	(Nasion, Sella), (Sella, Basion)
NSAr	(Nasion, Sella), (Sella, Articulare)
NSGn	(Nasion, Sella), (Sella, Gnathion)
ArGoMe	(Articulare, Gonion), (Menton, Gonion)
Mandibular pl to SN	(Gonion, Menton), (Sella, Nasion)
Gonial angle	(Articulare, Gonion), (Gonion, Menton)
U1 to SN	(mx1 root, mx1-incisor superius), (Sella, Nasion)
L1 to mandibular pl	(md1 root, md1-incisor inferius), (Menton, Gonion)
Interincisal angle	(mx1 root, mx1-incisor superius), (md1 root, md1-incisor inferius)
Occlusal pl to SN	(mx1-incisor superius, occlusal pl.), (Sella, Nasion)

**Table 3 diagnostics-13-03369-t003:** RMSE of each coordinate value of skeletal landmarks and its average error (mm).

	MRA	LASSO	RBFN	MLP	GBDT
Porion	1.19	0.67	7.01	4.23	24.76
Basion	2.44	1.35	7.25	4.11	1.87
Nasion	1.13	0.77	3.73	2.50	2.47
Orbitale	1.60	0.99	4.68	3.02	1.96
A point	2.99	1.62	7.57	4.13	3.23
Pogonion	3.71	2.43	12.77	8.78	1.85
B point	2.47	1.64	10.78	5.90	2.10
PNS	1.90	1.08	6.66	4.34	2.70
ANS	2.40	1.43	6.91	3.99	1.66
R1	2.49	1.03	7.01	3.89	1.95
R3	1.59	0.97	6.25	2.81	2.07
Articulare	1.33	0.77	6.39	3.85	2.34
Menton	3.07	1.80	12.72	7.62	2.24
mx1-incisor superius	2.89	1.66	9.61	3.80	2.29
mx1 root	3.38	1.88	8.16	4.44	2.33
md1-incisor inferius	3.13	1.56	9.32	3.74	2.43
md1 root	2.88	1.77	10.48	6.21	2.46
Occlusal plane	2.68	1.55	9.44	3.66	2.53
mx6 distal	3.01	1.53	7.78	4.03	2.48
mx6 root	2.79	1.66	7.11	5.01	2.88
md6 distal	3.10	1.56	8.35	3.85	2.95
md6 root	3.44	1.75	9.53	6.29	3.37
Gnathion	3.18	1.76	12.99	7.18	4.93
Gonion	2.20	1.39	10.22	5.89	2.68
Condylion	1.23	0.67	5.86	3.16	3.13
Average	2.49	1.41	8.34	4.66	3.43

RMSE, root-mean-square error; MRA, multiple regression analysis; LASSO, least absolute shrinkage and selection operator; RBFN, radial basis function network; MLP, multilayer perceptron; GBDT, gradient-boosted decision tree.

**Table 4 diagnostics-13-03369-t004:** RMSE of each skeletal linear parameter and its average error (mm) and accuracy (%).

	MRA	LASSO	RBFN	MLP	GBDT
ANS–Me	2.22	1.33	5.30	3.58	1.96
Ar–Go	2.54	1.31	4.11	2.43	1.60
Ar–Me	3.28	1.67	5.55	4.78	4.06
Cd–Go	3.00	1.36	4.57	3.01	3.59
Gn–Cd	2.72	1.79	5.65	4.11	2.64
Go–Me	2.37	1.56	4.11	3.19	1.68
N–ANS	1.43	0.92	2.57	2.12	1.34
N–Ba	2.53	1.52	4.81	3.74	1.93
N–Me	2.49	1.73	6.21	3.22	1.67
Pog–Go	2.29	1.65	4.38	3.28	2.08
S–Ba	2.20	1.19	2.95	2.57	1.98
S–Gn	2.32	1.80	7.57	3.39	2.35
Average	2.45	1.49	4.81	3.29	2.24
Accuracy (%)	96.39	97.87	93.29	95.33	96.73

RMSE, root-mean-square error; MRA, multiple regression analysis; LASSO, least absolute shrinkage and selection operator; RBFN, radial basis function network; MLP, multilayer perceptron; GBDT, gradient-boosted decision tree.

**Table 5 diagnostics-13-03369-t005:** RMSE of skeletal angular parameters and its average error (°) and accuracy (%).

	MRA	LASSO	RBFN	MLP	GBDT
A-B_plane	2.67	1.30	2.79	3.61	3.38
ANB	1.49	0.74	1.86	1.45	15.54
ArGoMe	3.65	1.57	5.60	3.66	1.99
Convexity	3.71	1.71	4.74	4.08	2.67
Gonial_angle	3.65	1.57	5.60	4.42	2.63
Interincisal_angle	8.25	3.75	9.23	6.48	1.80
L1_to_mandibular_pl	5.05	2.08	5.85	5.74	1.57
Mandibular_pl_to_SN	2.77	0.92	6.23	3.93	1.65
N-Pog_to_SN	1.61	0.69	5.47	1.98	1.41
NSAr	1.15	0.91	6.84	3.26	3.82
NSBa	1.31	0.74	6.36	5.64	4.07
NSGn	0.79	0.32	5.84	2.66	1.82
Nasal_floor_to_SN	1.58	0.70	4.63	2.52	2.38
Occlusal_pl_to_SN	25.77	11.69	16.06	14.75	2.01
SNA	2.03	1.03	5.47	2.50	1.12
SNB	1.13	0.59	5.31	2.49	1.64
U1_to_SN	6.60	2.73	8.22	6.33	1.88
Average	4.31	1.94	6.24	4.44	3.02
Accuracy (%)	88.21	94.45	82.75	85.35	63.42

RMSE, root-mean-square error; MRA, multiple regression analysis; LASSO, least absolute shrinkage and selection operator; RBFN, radial basis function network; MLP, multilayer perceptron; GBDT, gradient-boosted decision tree.

## Data Availability

The data presented in this study are available from the corresponding author upon request.

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
