# Peer review of "Validation of Machine Learning Models for Craniofacial Growth Prediction"

_diagnostics, 2023, doi:10.3390/diagnostics13213369_

Round 1

Reviewer 1 Report

Comments and Suggestions for Authors

This manuscript aims to identify the best model for predicting the accuracy from four modelling techniques (x3 ML, x1 traditional) using longitudinal lateral cephalometric radiographs to predict longitudinal craniofacial growth in the Japanese population. The manuscript is generally well written and contains an extensive Discussion. The use of diagrams are informative, but require minor refinements. The methods are relatively complex, require further revisions and the manuscript would benefit from a flowchart outlining the stages and techniques used, including the data age groups used so as to avoid any confusion. Please review your equations to highlight the repeated measure nature of your analysis. Some of your sections appear to be incorrectly titled and require review and amendments. Further comments and suggested edits are below:

Line 29: Please clarify to the reader what diagnosis in your first sentence “An accurate diagnosis is necessary…” (i.e. accurate diagnosis of what?)

Line 30: Please clarify to the reader what proportion of adult patients this relates to and provide the numbers in in your sentence “Although the proportion of adult patients has increased in recent years…” (i.e. proportion of what?)

Figure 2: This figure is very informative, but suggest thickening the lines and making the facial image grey rather than black to make the paths clearer for the reader.

Figure 3: Although this figure is also informative, suggest making arc lines thicker and the facial image grey rather than black so that the lines and arcs are prominent.

Title 2.3: The title for this section seems incorrect as there is more discussion than just the normalization of the data. Please review and amend accordingly.

Equations: your equations incorporate repeated measures but this is unclear to the reader. Please ensure you define all characters (e.g. i, j equations (1) and (2) etc). Please explain key elements of your equations (e.g. lambda in equation (3)). Please check this is done for all your equations.

Lines 173-175: Your statement “In the least squares method, the stepwise method evaluates the model using the adjusted R2 [31] and adopts the independent variables that provide the best evaluation.” is simplistic as the stepwise methods iteratively adds/removes significant/not significant independent variables and this will affect the adjusted-Rsquare. The stepwise method used dictates how this process is performed and can also be problematic if there are suppressor effects. Please expand on your explanation regarding the stepwise regression method of selection used and how you avoided the suppressor effect issue. For example, was it forward, backward, hierarchical etc stepwise? What level of p-value(s) was used for selection/elimination?

In your methods, under each method, you should include all your tuning parameters.

Lines 236-243: This section is under the heading of “2.3.4. Cross-validation” but relate to the performance metrics.

Section 2.3.5.: Further explanation and justification based on research is needed for your use of data for age 6 only and for using data for ages 6 to 7, 6 to 8, 6 to 9, 6 to 10 6 to 11, and 6 to 12 were used as input data.

Table 3: The value for Porion for GBDT is very high at 24.76. Please discuss this anomaly.

Table 5: Please explain why the values for  Occlusal_pl_to_SN (i.e. 25.77, 11.69, 16.06, 14.75) are very high for all models except GBDT, but for the GBDT only ANB is high (15.54).

Lines 407-409: Please clarify the following statements “This study included 39 independent variables and 163 lateral cephalometric radiographs. This study included 56 independent variables and 468 lateral cephalometric radiographs.”

Comments on the Quality of English Language

Please review some of the English as there are sentences that require clarification (see my comments). Further checking of the English grammar is required.

Author Response

Dear reviewer1

I will submit my reply as a Word file. 

Thank you very much.

Reviewer 2 Report

Comments and Suggestions for Authors

Introduction: Please mention the novelty of your study

Introduction: Please add new references.

Materials & Methods: Your sample size is small. What was the sample size in similar studies?

Discussion:  In the discussion section I would like to see a more profound discussion about the findings.

The study is well written and is well structured but the grammars and language should be checked again.

1.      One big question about this article is how researchers took another lateral cephalogram from children that it seems unethical.

2.      Discussion part should be more related and comprehensive

Author Response

Dear reviewer2

I will submit my reply as a Word file. 

Thank you very much.

Round 2

Reviewer 1 Report

Comments and Suggestions for Authors

Thank you for your revisions. There are only two minor outstanding issues that need addressing:

LINE 29: The English is still incorrect. Please amend to "Accurate diagnosis of a patient's tooth placement, bite, and craniofacial development is necessary...."

METHODS: Stepwise regression - please expand as an addition of a variable in a regression model will improve the R-square, even if it is by a very marginal amount. Please revise your explanation and see the cautionary issues (Smith, G. Step away from stepwise. J Big Data 5, 32 (2018). https://doi.org/10.1186/s40537-018-0143-6)

Comments on the Quality of English Language

Please amend the first sentence in your Introduction thank you.

Author Response

October 26, 2023

Dear reviewer1

I will submit my reply as a Word file. 

Thank you very much.

Sincerely,

Eungyeong Kim

Department of Orthodontics, School of Dentistry, Kanagawa Dental University,

Kanagawa 238-8580, Japan

Tel.: +81-46-822-8884

[email protected]
